# Lip Seal Strength and Tongue Pressure among Japanese Male Workers: Comparison of Different Age Groups

**DOI:** 10.3390/ijerph20042862

**Published:** 2023-02-06

**Authors:** Akira Minoura, Yoshiaki Ihara, Hirotaka Kato, Kouzou Murakami, Yoshio Watanabe, Kojiro Hirano, Yoshinori Ito, Akatsuki Kokaze

**Affiliations:** 1Department of Hygiene, Public Health and Preventive Medicine, School of Medicine, Showa University, Tokyo 142-8555, Japan; 2Division of Oral Functional Rehabilitation Medicine, Department of Special Needs Dentistry, School of Dentistry, Showa University, Tokyo 145-8515, Japan; 3Department of Radiology, Division of Radiation Oncology, School of Medicine, Showa University, Tokyo 142-8555, Japan; 4Division of Respiratory Medicine and Allergology, Department of Medicine, School of Medicine, Showa University, Tokyo 142-8555, Japan; 5Department of Otorhinolaryngology Head and Neck Surgery, School of Medicine, Showa University, Tokyo 142-8555, Japan

**Keywords:** Japanese, lip seal strength, tongue pressure, workers

## Abstract

Lip seal strength and tongue pressure are related to sarcopenia in older adults and are directly linked to the quality of life of workers after retirement. This study examined lip seal strength and tongue pressure among Japanese male workers by age. A self-administered questionnaire survey including alcohol consumption and smoking was conducted on 454 male workers. Height, weight, lip seal strength, and tongue pressure were also measured and then stratified by age (20s, 30s, 40s, 50s, and 60s and over). The mean (25th, 75th percentiles) lip seal strength and tongue pressure for all workers were 13.7 N (11.6, 16.4) and 41.7 kPa (35.2, 48.2), respectively. Both lip seal strength and tongue pressure were lowest in the 20s, at 12.1 N (9.6, 14.0) and 40.6 kPa (33.4, 47.6), respectively. The multiple regression analysis adjusted for smoking showed a significant positive association between lip seal strength and BMI for the 20s, 50s, and 60s and over, and a significant positive association between tongue pressure and BMI for the 30s, 40s, 50s, and 60s and over. To maintain oral health in older adults, it may be useful to measure workers’ lip seal strength and tongue pressure and intervene at an earlier stage.

## 1. Introduction

Various movements of the lips and the tongue are not only involved in multiple functions of oral motoring such as swallowing or chewing, but also activities of daily living and cognitive function [1,2]. Although lips and tongues are composed of multiple muscles, lip seal strength and tongue pressure play an important role in articulation, swallowing, and maintaining dentition [3,4]. For workers, oral health has an impact on not only their prevention of disease but also their performance at work [5]. However, some changes in the oral environment have few subjective symptoms, and regular measurement may lead to early disease prevention. Kugimiya et al. [6] demonstrated that lip seal strength has an aging effect and was not normally distributed of normally in men. Moreover, lip seal strength and tongue pressure were significantly correlated among Japanese [6]. As oral health may differ by age group, it is meaningful to consider the age group to promote effective checks and interventions by industrial doctors.

Although poor oral function is associated with various chronic diseases, there have been very few studies examined in healthy subjects, and studies from a preventive medicine perspective are currently insufficient. While malnutrition resulting from poor dietary intake can cause sarcopenia, it has been suggested that lip seal strength and tongue pressure are associated with nutrition-related sarcopenia in older adults and are directly related to the quality of life of workers after retirement [7,8]. The average life expectancy of Japanese men is about 81 years, which means that they will live longer in retirement than in the rest of the world [9]. Therefore, it is of great significance to measure lip seal strength and tongue pressure in Japanese men to maintain their quality of life after retirement. While the prevalence of sleep-disordered breathing related to oral health was 9% for men in the 1990s [10], it increased to 10–17% in the 2010s [11]. The causes of the increasing prevalence of sleep disorders over the years have not been identified and should be promptly investigated to improve the quality of life of workers.

In relation to oral motor skills or functions (e.g., evaluation of pronunciation and mastication), lip seal strength is utilized to evaluate maintenance or changes in oral health over time. It has been suggested that lip seal strength and tongue pressure are positively related to BMI, but due to the lack of data on healthy subjects, the effects of aging are unclear [12]. With the development of oral rehabilitation in recent years, assessing maintenance or changes in lip seal strength by age and its interventions could be expected to help in evaluating the lip seal strength of individuals and its interpretation for adults. The prevalence of sleep-disordered breathing in Japan is also reported to be 9% for men [13]. The prevalence of metabolic syndrome, which is a risk factor for sleep apnea syndrome (SAS), is also said to be about 20%. However, to our knowledge, no reports have been published on the effects on vocational drivers and shift workers in small- and medium-sized enterprises [14]. In these days of automation and mechanization in the service industry, there are many tasks that workers still have to operate and manage and preventing workers from turning their attention to them during the day is highly significant. Therefore, it is beneficial from the perspective of worker health care and prevention to determine the lip seal strength and tongue pressure of workers by age group.

Given this background, we hypothesized that lip seal strength and tongue pressure would differ by age group among workers. Thus, the purpose of this study was to examine lip seal strength and tongue pressure by age group among Japanese male workers.

## 2. Methods

### 2.1. Participants

The study participants were 455 male workers employed at 2 organizations in Japan. This survey was conducted from November 2021 to June 2022. Written informed consent was obtained from all participants prior to participation in the study, and we have estimated from the comorbidities that all participants were capable of measuring the lip seal strength and tongue pressure. Information on age, alcohol consumption (once a week or more, less than once a week), smoking (yes, no), and body mass index (BMI) were collected from all participants.

This study was approved by the Medical Ethics Committee of the School of Medicine, Showa University (approval No. 21-088-A).

### 2.2. Lip Seal Strength Measurements

In this study, a lip seal strength measuring digital device (Rippurukun, Shofu, Japan) was used to evaluate the worker’s lip seal strength (Figure 1). Among adults, the reliability of this device has been reported by some studies in Japan [10,11]. A floss (dental floss for measuring lip seal strength, Johnson and Johnson, New Brunswick, NJ, USA) measuring about 30 cm long was tied (in the shape of a ring), and a specific button for measuring (Rippurubotan, Shofu, Kyoto, Japan) was attached to the tip of Rippurukun [11]. The lip seal strength measuring digital device, which has an oral screen-like shape, was placed in the oral vestibule [11]. During measurements, the participants sat with the head position parallel to both the floor and the Frankfort plane. The workers had three instructions before the measurement to eliminate measurement errors. (1) Resist traction using only the muscle strength of the lips (not using the strength of the tooth or tongue). (2) Do not resist traction using sucking force because if the intraoral pressure becomes negative pressure, accurate measurement cannot be performed. (3) Do not swing the neck and bend the body backward or forward while resisting. During measurements, the measurer stood in front of the workers and measured the lip seal strength. The workers were instructed to close their lips after confirming that they had placed the specific button for measuring between the lower and upper anterior tooth regions, and the button was positioned at the center of the dentition. At the same time, the button was pulled slowly in the horizontal direction while maintaining its position. Measurements were repeated three times while ensuring that a break was taken, and the mean value was defined as the participant’s lip seal strength (N). Since results of repeated lip seal strength are largely dependent on the person measuring technique for the strength, all measurements of the strength were performed entirely by a dentist with expertise in measurements. To reduce potential bias, all workers were measured by the same dentist, as lip seal strength could vary depending on the dentist’s instructions.

### 2.3. Tongue Pressure Measurements

In this study, a special device for measuring tongue pressure [kPa] (TPM-01, JMS, Hiroshima, Japan) was used to measure the worker’s tongue pressure (Figure 2). For all participants, tongue pressure was measured by a special device using an apparatus with a balloon-type oral probe. Firstly, the probe was placed between the center of the tongue and the upper incisors on the hard palate. Secondly, the workers kept closing their lips and pressed them as much as possible as the tongue against the hard palate. The measurement took several seconds, and the momentary maximum air pressure of the probe was taken as the tongue pressure. Tongue pressure was measured three times repeatedly, and the mean value was recorded as the worker’s value in the analysis. Each measurement was made with at least 1 min of rest between measurements [15]. Since results of repeated tongue pressure are largely dependent on the person measuring the tongue pressure, all measurements of tongue pressure were performed entirely by a dentist with expertise in measurements. To reduce potential bias, all workers were measured by the same dentist, as tongue pressure could vary depending on the dentist’s instructions.

### 2.4. Statistical Analysis

All statistical analyses were conducted using STATA 12.0 (Stata Corp, College Station, TX, USA) and JMP 16.2 (SAS Institute Inc., Cary, NC, USA). A Shapiro–Wilk test was performed on all continuous variables to confirm normal distribution. For the continuous variables, analysis of variance was performed for a normally distributed continuous variable, and Kruskal–Wallis test was performed for continuous variables not normally distributed. Chi-squared tests were performed for alcohol consumption and smoking. Comparisons of data between the 20s and the 30s and over were analyzed by Dunnett’s test for continuous variables with 20s as the control group. To examine the relationship between lip seal strength and tongue pressure, Spearman’s correlation analysis was conducted for coefficient measurements, including BMI, and multiple regression analysis adjusted for BMI and smoking. Our study followed the STROBE (strengthening the reporting of observational studies in epidemiology) guidelines for a cross-sectional study. All analyses were carried out with 95% confidence intervals. *p*-values < 0.05 (two-tailed) were considered statistically significant.

## 3. Results

After excluding one worker for whom tongue pressure could not be measured, a total of 454 workers were included in the analysis. The characteristics of the study participants are shown in Table 1. Of the continuous variables, only tongue pressure was normally distributed.

Among all workers, the median (25th, 75th percentiles) lip seal strength was 13.7 N (11.6, 16.4), and the median tongue pressure was 41.7 kPa (35.2, 48.2). The median age was 47.0 (33.0, 55.0). Among 20s workers (*n* = 91), the median lip seal strength was 12.1 N (9.5, 14.0), and the median tongue pressure was 40.6 kPa (33.4, 47.6). Among 30s workers (*n* = 64), the median lip seal strength was 13.0 N (11.6, 15.7), and the median tongue pressure was 45.0 kPa (37.7, 50.1). Among 40s workers (*n* = 92), the median lip seal strength was 14.6 N (12.4, 17.3), and the median tongue pressure was 44.4 kPa (35.3, 50.2). Among 50s workers (*n* = 147), the median lip seal strength was 14.3 N (12.2, 17.0), and the median tongue pressure was 41.7 kPa (36.7, 47.4). Among 60s and over workers (*n* = 60), the median lip seal strength was 14.9 N (12.0, 18.0), and the median tongue pressure was 41.0 kPa (34.9, 46.6). The results of the age comparison analyzed by Dunnett’s test by age with 20s as the control group revealed that values were significantly higher for the 20s. BMI was lowest among those in their 20s and highest among those in their 60s and over. No significant age-specific associations were found for alcohol consumption. Regarding smoking rates, those in their 20s had the lowest rates and those in their 40s had the highest rates. Histograms of lip seal strength and tongue pressure are shown in Figure 3, and the results of Spearman’s correlation analysis are shown in Table 2. 

The mode of lip seal strength was 12.0 to 13.0, and the mode of tongue pressure was 38.0 to 40.0. The lip seal strength followed a normal distribution, but the tongue pressure did not follow a normal distribution. There was a significant correlation between lip seal strength and tongue pressure in all age groups. Especially, the correlation coefficient between lip seal strength and tongue pressure in the 20s was 0.50, which was higher than in other age groups. The correlation coefficients between lip seal strength and tongue pressure for those in their 30s and over were 0.29 for those in their 30s, 0.36 for those in their 40s, 0.38 for those in their 50s, and 0.39 for those in their 60s and over. Lip seal strength was significantly associated with BMI in the 20s, 50s, and 60s and over. The correlation coefficients between lip seal strength and BMI for those in their 30s and over were 0.24 for those in their 50s, and 0.35 for those in their 60s and over. Tongue pressure was significantly correlated with BMI in all age groups except the 20s. The correlation coefficients between lip seal strength and tongue pressure for those in their 30s and over were 0.31 for those in their 30s, 0.27 for those in their 40s, 0.30 for those in their 50s, and 0.30 for those in their 60s and over. The results of the Shapiro–Wilk test indicated that only lip seal strength had a normal distribution. The results of multiple regression analyses for lip seal strength and tongue pressure by age group are shown in Table 3. After adjusting for smoking, lip seal strength was positively associated with BMI in the 20s, 50s, and 60s and over, while tongue pressure was positively associated with BMI in the 30s, 40s, 50s, and 60s and over. Neither lip closure force nor tongue pressure showed a significant association with smoking.

## 4. Discussion

The results of this study suggest that the tendencies of lip seal strength and tongue pressure of workers differ among those 20s (young adults), the age of middle, and the elderly. Both the lip seal strength and tongue pressure of male workers in their 20s were lower than those in their 30s, 40s, 50s, 60s, and over. Especially, in the 20s workers, lip seal strength was significantly lower than that of any other age group in the 30s and above. Tongue pressure was lower in the 20s than in the 30s but tended to decrease gradually in the 40s and over. The results also suggested that tongue pressure was significantly positively associated with BMI among Japanese workers in their 20s, 50s, and 60s and older, even after adjusting for smoking. Moreover, tongue pressure was significantly positively associated with BMI in all age groups except the 20s. Therefore, the background of this age distribution of lip seal strength and tongue pressure may be due to a combination of both lifestyle influences specific to young people and the effects of aging among male workers.

To our knowledge, this is the first study to measure and examine the association between lip seal strength and tongue pressure among a wide age range of Japanese male workers. In previous studies, lip seal strength and tongue pressure have been shown to have a significant impact on the deterioration of oral health in older adults [8]. As lip seal strength can be improved through lip seal resistance training, early interventions for oral health maintenance in older adults can be implemented if a decline in lip seal strength is identified at a younger age [16]. Early intervention to prevent loss of lip seal strength and tongue pressure may minimize the impact on work performance and the cost of medical treatment or oral training. The results of this study also suggested that eating habits, such as the number of times a meal is chewed, may differ between young people and middle-aged and older adults.

Although the effects on health until an elderly age are limited, it can be defined that lip seal strength and tongue pressure during working life affect their life after retirement. Swallowing exercises that include forward and bilateral pulling can help increase the strength and mobility of the lip seal strength over time [17]. In addition, poor oral health and decreased tongue pressure are independent risk factors for salivary bacterial levels [18]. Decreasing lip seal strength has been demonstrated to be one of the factors that lead to a deterioration of complex dental disorders or oral functions, but there were few adult studies on lip seal strength in Japan [1]. Given this background, the cut-off value of lip seal strength for judging oral disorders has not been determined because the distribution of lip seal strength remains unclear at least in healthy Japanese people. Therefore, the purpose of this study is designed to assess the lip seal strength and provide characteristics that can form part of the evidence for determining the cut-off of lip seal strength among Japanese male workers. In addition, a previous study reported an association between tongue pressure and the motor function of the lips with chewing, and lip seal strength was significantly associated with and masticatory performance and tongue pressure [8]. The results of this study suggested that there were significant differences in lip seal strength at least among male workers by age, suggesting that lifestyle and age-related changes in oral health may be related to sleepiness in the daytime.

The relationship between lip closure force and tongue pressure and BMI is discussed. Recent research suggests that BMI is associated with lip seal strength [12,19], and lip seal resistance training can increase lip seal strength in young adults [16]. Obese patients may experience changes to their stomatognathic system, which includes the lips [4], and exercises such as lip closure can help increase lip strength and mobility over time, potentially aiding swallowing [5]. Recent studies have shown a relationship between maximum tongue pressure and BMI [20,21], and the influence of height, weight, and BMI on axial tongue muscle strength [22]. Moreover, tongue pressure has also been related to dental arch length and width, BMI, and weight [22,23]. Therefore, it is possible that the force used to close the lips and the pressure on the tongue are intermediate factors in chronic diseases that are thought to be caused by being overweight/obese. Regardless of the degree, sleepiness during working hours can have a significant negative impact on job performance, and SAS is known to be a major occupational health issue that directly affects workers’ performance on the job [24,25]. As the proportion of older adults in Japan is rapidly increasing, the Japanese government is promoting employment among the aged. Promoting the employment of the elderly has the potential to contribute to improving the health of the elderly and securing the labor force, but at the same time, efforts must be made to prevent work-related accidents and illnesses. As the depression of the tongue root due to aging is known to promote SAS, examining age-related changes in lip seal strength and tongue pressure among workers may help in the prevention of SAS. For drivers in particular, the early detection and prevention of SAS are important because SAS involves a risk that can directly affect the lives of the workers themselves and their customers. It is possible to improve lip seal strength through training and understanding the lip seal strength of workers is thought to lead to greater maintenance of the labor force [26]. Moreover, metabolic syndrome, a risk factor for SAS, is reported to have a prevalence of about 20%, but there are no reports on gender differences, severity, and effects in work systems, especially for occupational drivers and shift workers in small and medium-sized enterprises. On the other hand, the results of this study indicate that prevention of METs may not necessarily prevent sleep disorders because the association of BMI with lip seal strength and tongue pressure varies with age among workers. The results of this study may allow a more in-depth examination of the relationship between lip seal strength, tongue pressure, and sleepiness in Japanese workers. While dental checkups are mandatory for children in Japan, dental checkups for workers including measurements of lip seal strength may contribute to preventing or improving sleep disorders.

The main strength of this study is its measurements of lip seal strength and tongue pressure in male workers ranging in age from their 20s to their 60s and over. Measurement of lip seal strength and tongue pressure in many healthy subjects was unprecedented, and the results of this study were likely to provide important insights for further preventive medicine studies using oral health checkups including lip seal strength and tongue pressure measurements. On the other hand, this study did have some limitations. First, we could not determine causal relationships because this was a cross-sectional study. It is hoped that future research will advance longitudinal studies and promote causal inferences to further elucidate the relationship between the oral environment and sleepiness among workers. Second, due to the absence of female workers in this study, gender differences in lip seal strength and tongue pressure could not be investigated, and some studies in Japan have indicated gender differences in the magnitude and directional specificity of lip seal strength produced during pursing-like lip-closing movements in healthy young adults [27,28]. As previous studies have reported gender differences in lip seal strength, further investigations of gender differences in occupational health are needed. While this study has secured more male subjects than many previous studies, it is necessary to examine more regions and workforces for generalizability in future studies. In Japan, women’s participation in the workforce has been slow in some industries, but we believe that future studies will further examine gender differences in the workforce. Third, in this study, individual data on BMI, alcohol consumption, and smoking, which significantly affect lip seal strength and tongue pressure, were obtained, but data on other lifestyle factors (such as eating speed, diet from a nutritional perspective, and exercise) were not. It would be beneficial to obtain these data in order to consider specific measures to prevent sleep disorders. Finally, because this study was conducted during the COVID-19 pandemic, it is possible that changes in the lifestyles of the participants due to infection prevention measures may have affected the oral function measurements. For example, some workers may have drastically altered their eating and exercise habits due to behavioral restrictions to prevent the spread of COVID-19 infectious diseases. Therefore, further examinations of lip seal strength and tongue pressure are needed at the end of the COVID-19 pandemic. In other words, the oral environment data obtained in this study were valuable because they were conducted on a large number of male workers during the COVID-19 pandemic.

## 5. Conclusions

In conclusion, the present findings suggest that tongue pressure and lip seal strength are lower among young workers than among those aged 30 years and older. It is possible that lip seal strength and tongue pressure are deteriorating in young workers because of changes in eating habits, so this relationship should be investigated in future studies. At the same time, to maintain oral health in older adults, it may be useful to measure workers’ lip seal strength and tongue pressure and intervene at an earlier stage.

## Figures and Tables

**Figure 1 ijerph-20-02862-f001:**
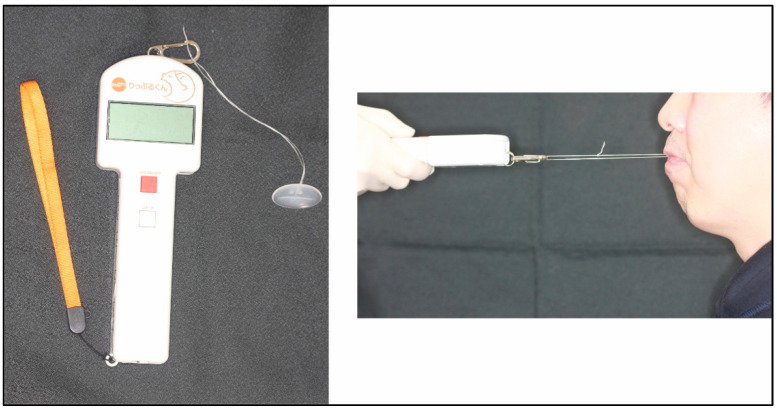
Measurement of lip seal strength.

**Figure 2 ijerph-20-02862-f002:**
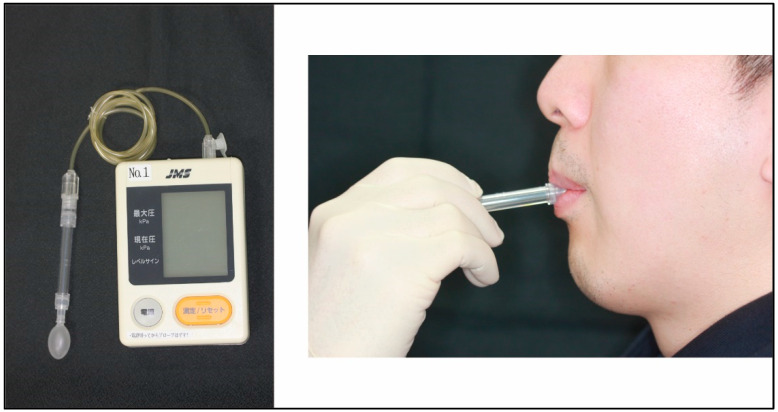
Measurement of tongue pressure. This device displays current and maximum pressures.

**Figure 3 ijerph-20-02862-f003:**
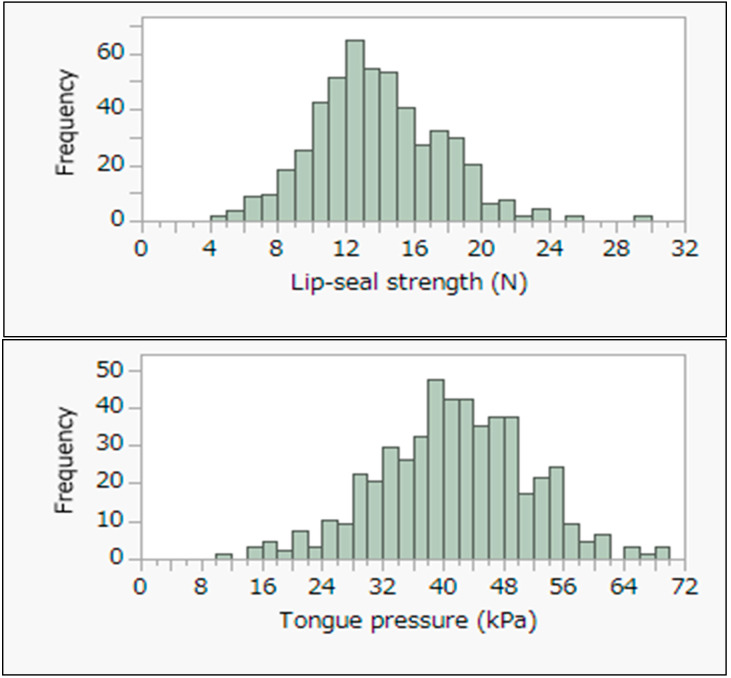
Histograms of lip seal strength and tongue pressure in workers (N = 454).

**Table 1 ijerph-20-02862-t001:** Characteristics of the study participants by age group.

	Total	20s	30s	40s	50s	60s and Over	^(1)^*p*-Value
	(N = 454)	(*n* = 91)	(*n* = 64)	(*n* = 92)	(*n* = 147)	(*n* = 60)	
**Lip-seal strength**	13.7 (11.6, 16.4)	12.1 (9.6, 14.0)	13.0 (11.6, 15.7) *	14.6 (12.4, 17.3) *	14.3 (12.2, 17.0) *	14.9 (12.0, 18.0) *	<0.001
**Tongue pressure**	41.9 (9.8)	39.8 (10.1)	43.3 (11.5) *	43.5 (9.8)	42.1 (8.4)	40.4 (10.1)	0.052
**Age**	47.0 (33.0, 55.0)						
**BMI**	24.1 (21.7, 27.1)	22.0 (19.8, 24.5)	24.0 (22.2, 28.3) *	24.7 (22.3, 27.4) *	24.3 (22.3, 27.4) *	25.2 (23.0, 28.7) *	<0.001
**Alcohol use**	269 (59.3)	49 (53.9)	35 (54.7)	59 (64.1)	84 (57.1)	42 (70.0)	0.223
**Smoking**	169 (37.2)	25 (27.5)	23 (35.9)	48 (52.2)	53 (36.1)	20 (33.3)	0.011

Except where indicated *n* (%), tongue pressure is mean (standard deviation), and the other values are median (25th, 75th percentile). BMI: Body mass index. ^(1)^ For continuous variables, analysis of variance was performed for tongue pressure, and Kruskal-Wallis test was performed for lip seal strength and BMI. For dichotomous variables, chi-square test was performed. * The results of the Dunnett’s test by age, using 20s as controls, were significantly higher.

**Table 2 ijerph-20-02862-t002:** Correlation between lip-seal strength, tongue pressure, and BMI by age group.

	20s		30s		40s		50s		60s and Over	
	(*n* = 91)		(*n* = 64)		(*n* = 92)		(*n* = 147)		(*n* = 60)	
**Lip-seal strength**										
✕	0.50	*	0.29	*	0.36	*	0.38	*	0.39	*
**Tongue pressure**										
**Lip-seal strength**										
✕	0.23	*	0.24		0.20		0.24	*	0.35	*
**BMI**										
**Tongue pressure**										
✕	0.12		0.31	*	0.27	*	0.30	*	0.30	*
**BMI**										

Values are correlation coefficient. BMI: Body mass index. * *p* < 0.05.

**Table 3 ijerph-20-02862-t003:** Multiple regression analyses for lip-seal strength and tongue pressure by age group.

		20s		30s		40s		50s		60s and Over	
		(*n* = 91)		(*n* = 64)		(*n* = 92)		(*n* = 147)		(*n* = 60)	
**Lip-seal strength**											
	BMI	0.18	*	0.22		0.15		0.19	*	0.32	*
	Smoking	−0.29		0.26		−0.38		−0.25		0.14	
**Tongue pressure**											
	BMI	0.28		0.75	*	0.64	*	0.59	*	0.76	*
	Smoking	−1.27		−0.49		0.38		−0.01		0.03	

Values are standardized beta. BMI: Body mass index. * *p* < 0.05.

## Data Availability

In this study, the data are not available in a public repository but are available from the corresponding author upon reasonable request (minoaki@med.showa-u.ac.jp).

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
