# Peer review of "Lip Seal Strength and Tongue Pressure among Japanese Male Workers: Comparison of Different Age Groups"

_ijerph, 2023, doi:10.3390/ijerph20042862_

Round 1
Reviewer 1 Report
The objective of this article was to examine the lip seal strength and the tongue pressure by age group among Japanese male workers. The justification of the study needs to be better described. The choice of the statistical tests should be better elucidated. The major issues that should be addressed are detailed:
Introduction
· The introduction feels a little disjointed. Could you please revise
· In the introduction Section the authors should emphasize more the role of the lip seal strength and tongue pressure during each function of mastication, swallowing and speaking.
· Authors might define the oral health status and its link to the lip and tongue pressure.
· The measurements of the lip seal strength and tongue pressure in male should be better justified.
· You mentioned the following sentence: “It has been suggested that lip seal strength and tongue pressure are related to sarcopenia in older adults and are directly linked to the quality of life of workers after retirement”. Please Add references and explain how it is directly linked to the quality of life of workers after retirement.
Materials and methods
· The design of the study might be mentioned
· The sampling method of the 455 male workers need to be better described. Please also add the rate of study participation. Was the sample size calculated ?
· What were the inclusion and excluded criteria for this study
· The method used to describe the measurements is well presented; however, it needs to be illustrated in photos as a case. Please, add illustrative photos about the digital device that measure the lip steals. Also add illustrative photos about the special device for measuring tongue pressure
· Please enumerate the effort addressed to reduced potential bias
Statistical analyses
Were variables normally distributed? Please also add the statistical tests used to assess normality distribution of continuous variables in the statistical analyses section and not in the results. Why the values of Percentiles were reported for lip seal strength, tongue pressure, age and BMI instead of mean and Standard-deviation, particularly that parametric tests (ANOVA) were used. Please clarify this point and I think that the mean values and Std Deviation should be also added in addition to median / percentiles.
Discussion
In the discussion section, the impact of a poor seal strength and tong pressure might also be fully discussed.
The role of BMI in this correlation need to be better addressed.
The Discussion section should be in line with the introduction.
In the Limitations section, please discuss generalizability of the results.
Author Response
Thank you for agreeing to review our manuscript. By revising from the introduction to the conclusion, we were able to improve our research.

Reviewer 2 Report
Well designed study however I feel conclusion need to be improved as lip seal strength and tongue pressure are decreasing with age, it can not be predicted as further longitudinal studies are performed. So with minor modification in conclusion, study can be improved.
Author Response
Thank you for agreeing to review our manuscript. By revising the conclusion, we were able to improve our research.
Round 2
Reviewer 1 Report
The authors have responded clearly and adequately to the questions.